# How to Catch the Ball: Fullerene Binding to the Corannulene Pincer

**DOI:** 10.3390/molecules27123838

**Published:** 2022-06-15

**Authors:** Filipe Menezes, Grzegorz Maria Popowicz

**Affiliations:** Institute of Structural Biology, Helmholtz Zentrum Muenchen, Ingolstaedter Landstr. 1, 85764 Neuherberg, Germany

**Keywords:** semi-empirical, buckycatcher–fullerene, π stacking, Born–Oppenheimer molecular dynamics, thermodynamics of binding

## Abstract

The corannulene pincer (also known in the literature as the buckycatcher) is a fascinating system that may encapsulate, among other molecules, the $C_{60}$ and $C_{70}$ fullerenes. These complexes are held together by strong π-stacking interactions. Although these are quantum mechanical effects, their description by quantum chemical methods has proved very hard. We used three semi-empirical methods, PM6-D3H4X, PM6-D3H+ and GFN2-xTB, to model the interactions. Binding to fullerenes was extended to all open conformations of the buckycatcher, and with the proper choice of solvation model and partition functions, we obtained Gibbs free energies of binding that deviated by 1.0–1.5 kcal/mol from the experimental data. Adding three-body dispersion to PM6-D3H+ led to even better agreement. These results agree better with the experimental data than calculations using higher-level methods at a significantly lower fraction of the computational cost. Furthermore, the formation of adducts with $C_{60}$ was studied using dynamical simulations, which helped to build a more complete picture of the behavior of the corannulene pincer with fullerenes. We also investigated the use of exchange-binding models to recover more information on this system in solution. Though the final Gibbs free energies in solution were worsened, gas-phase enthalpies and entropies better mirrored the experimental data.

## 1. Introduction

Since the synthesis of the first buckybowls by Sygula et al. [1], these species have attracted much attention in the scientific community. These derivatives of corannulene resemble a clothespin and are sufficiently flexible to capture a wide range of molecules. Of particular interest is the binding to fullerenes $C_{60}$ and $C_{70}$. The key factor for this interaction is the compatibility of the surfaces of the buckycatcher with the (almost) spherical surface of the buckminsterfullerenes. This permits strong π stacking interactions [2] that keep the structure intact [1]. From as theoretical point of view, properly balancing these interactions prove to be quite a challenging task with many contradictory results.

Using the Minnesota functionals, M06-L and M06−2X [3], Zhao and Truhlar estimated the binding energy of $C_{60}$ to corannulene pincers to be approximately 26.4 kcal/mol. Further inclusion of dispersion corrections strengthens the interaction so that with the M06-L + TS-vdW method binding was estimated to lie around 46.5 kcal/mol [4]. This is approximately 20 kcal/mol apart from the previous values. Employing dispersion corrected density functional theory (DFT), Grimme [5] estimated the binding energies of 35.0 and 36.9 kcal/mol for $C_{60}@C_{60}H_{28}$ and 36.1 and 38.6 kcal/mol for $C_{70}@C_{60}H_{28}$. In other studies, values as large as 43.1 kcal/mol were also reported [6]. Later on, Sure and Grimme back-calculated the binding energies for these complexes from experimental data [7], for which they obtained the values of 28.4 ± 0.6 kcal/mol for $C_{60}@C_{60}H_{28}$ and 29.8 ± 0.7 kcal/mol for $C_{70}@C_{60}H_{28}$. The latter are in good agreement with diffusion Monte Carlo (DMC) data, which estimates a binding of 26.0 ± 2.0 kcal/mol [8]. Within the symmetry-adapted perturbation theory (SAPT) framework, Carter-Fenk et al. [9] found slightly higher binding energies for these complexes, respectively, 31.96 and 31.68 kcal/mol. More recently, Ballesteros et al. [10] calculated the binding of $C_{60}$ to the corannulene pincer using high-level ab initio methods such as MP2 and DL-PNO-CCSD(T). While the former clearly overestimates binding (78.3 kcal/mol), the latter predicts interactions stronger than the reference data by 15 kcal/mol. The DL-PNO-CCSD (T) results might be negatively influenced by the MP2 contribution in the calculation of the complete basis set (CBS) energies as well as in the method itself. None of the works thus far mentioned considered the effects of other conformers of the buckycatcher for the complexation of fullerenes. Furthermore, no information whatsoever was given on the path leading to the formation of these adducts.

The isolated buckycatcher was also subject to DFT studies. Mück-Lichtenfeld et al. [6] as well as Denis and Iribarne [11] have considered four possible conformers that describe this species. The results reported were strongly dependent on the selected density functional. For instance, M06−2X, PBE-D2 and PBE-D3 disfavor the conformer with strong intramolecular π stacking. On the other hand, B97-D predicts this species to be, by far, the most stable in gas. In our work [12], we verified that the exact same problem was observed when using semi-empirical methods. Fortunately, this conformer is not of relevance in solution due to the fact of its reduced solvent accessible surface area [12]. It may thus be safely neglected. The interconversion between conformers has also been tackled in the literature [6,11]. The barriers obtained with DFT are between 8 [11] and 11 kcal/mol [6]. Using semi-empirical methods [12], barriers of approximately 9 kcal/mol were obtained. Consequently, all methods are consistent.

In the current work, we re-evaluated the Gibbs free energies for the complexation of fullerenes to the corannulene pincer at three semi-empirical levels, using conformer averaged statistical mechanics. We tested and compared several solvation models and their influence on the results. Though already discussed in statistical terms [7], we show concrete examples that demonstrate how beneficial it would be for PM6-D3H+ to include three-body dispersion corrections.

## 2. Results

### 2.1. Systems Studied and Nomenclature

We identified four main conformers for the corannulene pincer [12]: *ii* for in-in; *ee* for ex-ex; *ie*. Figure 1 shows these species complexed with $C_{60}$. A fourth conformer, *iet* (t stands for tight), was identical to *ie*; however, the corannulene units were closely π stacked. Contrary to the other three (open) species, *iet* itself cannot encapsulate fullerenes. Consequently, this species was disregarded in this study. This is supported by our previous studies on the system [12].

A measure of how wide apart the corannulene units are in the free or complexed buckycatcher is defined by $R_{P}$ [3,11] (Figure 1 and Table 1). We observed that GFN2-xTB structures were up to 1–1.5 Å wider than PM6 structures. Nevertheless, the qualitative picture was identical for all methods and even the point group symmetries matched well the S12L data [13].

### 2.2. Binding Energies

Binding energies with dispersion-corrected PM6 methods for conformer *ii* were closer to the reference values than GFN2-xTB, a fact that was already observed before for PM6-D3H4X (supporting materials in [14]). Using GFN2-xTB-optimized geometries led to even larger deviations with respect to reference values than when S12L geometries were employed. This was because the binding at GFN2-xTB’s equilibrium geometry caged the fullerenes tighter, a consequence of the strengthened interaction with respect to the S12L case (there was a change of approximately 0.3 Å in $R_{P}$). Compared to GFN2-xTB, dispersion corrected PM6 methods expect the catcher to contract further when it is free. This intensified the intramolecular interactions in the free species, which consequently lowered the energy of the reagents with respect to the products. The net effect on binding (with respect to the S12L geometries) brought the binding energies closer to the reference values [7].

Despite the similarity of the geometries obtained with dispersion-corrected PM6 methods, there was a difference of approximately 3 kcal/mol in the binding energies between the two methods. We traced this back to repulsive dispersion interactions, which were absent in D3H+. The PM6-D3H * (PM6-D3H+ with three-body dispersion) binding energies were in much better agreement with the reference data: 27.906 kcal/mol for $C_{60}$ and 27.707 kcal/mol for $C_{70}$. On the other hand, PM6-D3H4X did not seem to benefit from three-body dispersion, opposing the observations of Sure and Grimme [7]. Careful analysis of the data showed that the values they calculated differed by up to 4.4 kcal/mol with respect to ours, which accounted for the different findings. In the same work, PM6-D3H+’s binding energies were also calculated, and these differed by up to 1.6 kcal/mol with respect to our values. As the data in Table 2 shows, the observed differences at the PM6-D3H+ level can be easily justified with dissimilar geometries. For PM6-D3H4X, the difference was larger by a further 2.4 kcal/mol, which with the lack of a proper analysis, we assumed it resulted from small variations in the parameters used.

For comparison purposes we also calculated the binding energies with noncorrected PM6, for which we obtained the values of 3.684 and 3.905 kcal/mol for $C_{60}$ and $C_{70}$. Although Lennard-Jones-like terms were present in PM6 to account for nonbonded interactions, the respective binding energies were very far off any of the other values. This gave us some freedom in interpreting the lower weight that the dispersion terms had in PM6-D3H4X. More than just avoiding double counting of dispersion contributions, rather we saw it as an indirect inclusion of repulsive (three-body) dispersion. This allowed for a systematization of the two-body dispersion-corrected PM6 methods we discuss here.

Comparing the PM6-D3H+ results against PM6 showed that the two-body dispersion contributions summed up to approximately 30 kcal/mol. Three-body dispersion accounted for approximately 10% of the two-body terms. Though we did not perform other checks for consistency, the trend in the results indicates that higher-order terms should be negligible for the systems we studied.

When studying the effect of the curvature of the pincer’s arms on the binding energy, all methods were extremely well-behaved and easy to rationalize. Bending each of the pincer’s arms outwards decreased the binding by approximately 6–8 kcal/mol. Furthermore, no synergistic effect was observed, meaning that the effect of each arm to binding was practically additive.

### 2.3. Thermodynamics of Pincer–Fullerene Complexes in Gas and in Toluene

Table 3 presents the main thermodynamic data for the formation of adducts of fullerenes with the buckycatcher at 300 K in toluene. The respective gas phase data are found in Table 4. Both tables show conformer-averaged statistics. Conformer-specific data are provided in the Appendix A. Expectedly, conformational effects strongly benefited the results.

The temperature dependence of gas-phase enthalpies and entropies was very weak, and a single value was almost sufficient for describing the quantities over a large range of temperatures. Nevertheless, the slope of the enthalpy plots was negative, in agreement with experimental observations [15]. With the harmonic oscillator partition function, the $\Delta C_{P}$s we calculated were all positive. Irrespective of the method, the calculated gas-phase $\Delta C_{P}$s were one order of magnitude smaller than the experimental ones. Though the accuracy of the calculated frequencies might partly account for these observations, we are inclined to assign the largest deviations to (lacking) solvation effects [15]. Nevertheless, using the Grimme partition function led to $\Delta C_{P}$s with the same signal as the experimental values.

Irrespective of the conformer or method considered, entropy penalties for the formation of the adduct with fullerenes were always uniformly very large with values ofapproximately −50 cal/(K.mol). This agrees well with previous calculations [5]. Using only the harmonic oscillator partition function led to changes in entropy higher by 10 cal/(K.mol). Experimental data in toluene yields values close to zero and slightly positive [15]. We conclude, as already suggested [15], that this is strong evidence for the direct involvement of the solvent in the reaction.

Data on Gibbs free energies show two main sets, separated by approximately 10 kcal/mol. These differences result exclusively from the binding energies, since the entropy changes were all identical. Denis and Iribarne [11] calculated gas-phase Gibbs free energies at 298 K and obtained, with the M06−2X functional, a value close to our PM6-D3H4X data (−14.3 kcal/mol). With the B97-D functional, which strongly overestimates binding energies (compare, for instance, the binding energies available in [11] with the reference values [7]), the same authors obtained a change in the Gibbs free energy in gas of −29.7 kcal/mol, which is closer to the GFN2-xTB data. We note that Denis and Iribarne calculated conformer-specific data, whereas, here, we present conformer-averaged thermodynamics. A direct comparison with the literature values must take place with data we provide in the Appendix A.

In agreement with other theoretical data [5], adding the effects of solvent raised all Gibbs free energies. The ALPB model was, overall, quite penalizing, with an estimated contribution of 10 kcal/mol. Such a penalty is, however, still insufficient to bring GFN2-xTB estimates close enough to the experimental data. When used with PM6 methods, the results improved significantly. Nevertheless, ALPB’s penalizing nature became obvious. Note that we assigned, in this case, most deviations to the solvation model because PM6–D3H4X’s binding energy was quite accurate, and rovibrational contributions to the enthalpy and entropy agree with higher-level calculations. Observed differences might, however, result from using ALPB’s parametrization for benzene and not specific parameters for toluene. The solvent contribution from COSMO was more modest, and the impact was only approximately 2 kcal/mol in total. All the calculated values for the affinities were thus too low, and we observed that this model was also unable to reproduce the experimental data in solution.

In an earlier study with the COSMO-RS model [16,17,18], Grimme [5] determined the solvation Gibbs free energies of 6.5 and 7.3 kcal/mol for the formation of the inclusion complexes of $C_{60}$ and $C_{70}$ with *ii*. The COSMO-RS solvation energies were calculated in a conceptually different way than what was conducted in any of the other two models discussed thus far. In a summarized way, the contribution from the solvent in the former was estimated by the interaction of molecular σ profiles which, to some extent, encode the nature of interactions at interplay. In this sense, it is quite elucidative to compare the results obtained when COSMO-RS corrections are used (rows ${\Delta G}^{METHOD/COSMO-RS}$ in Table 3), even if the geometries used by Grimme in such an estimation certainly differed from ours, and there were only data for one of the conformers in the literature. In the Appendix A, we estimate errors that might stem from these approximations. The COSMO-RS correction in combination with the PM6-D3H4X gas Gibbs free energies yielded exceptionally good binding affinities in toluene for either of the fullerenes, which deviated by 1.25–1.50 kcal/mol from the experimental data. Despite the significant error cancellation that benefits anyway from any quantum chemical method, such an agreement with experimental data is, in our opinion, quite remarkable for such a low-level method: these results outperformed higher-level calculations both in calculation time and in quality. As already discussed, PM6-D3H+ overestimated the binding energy. With the addition of the conformer-averaged Axilrod–Teller–Muto corrections to PM6-D3H+, we obtained Gibbs free energies in toluene of −4.794 and −4.778 kcal/mol for, respectively, $C_{60}$ and $C_{70}$. These values will surely slightly change once a proper parametrization is introduced. The potential benefit for the method is, however, undeniable.

### 2.4. Conformational Equilibria in the Presence of Fullerenes

Though binding energies at the GFN2-xTB level disagreed with the results obtained with dispersion-corrected PM6 methods, the relative energies of the adducts between the buckycatcher and fullerenes were identical for all the methods. We used, therefore, GFN2-xTB to study the conformational equilibria of the buckycatcher in the presence of a fullerene. In Figure 2, we present the energy surface connecting the several conformers of the buckycatcher with and without $C_{60}$ within the pincer’s arms. Furthermore, the differences in energy match the relative chemical potentials in gas ($\mu \left[C_{60}@ii\right]$ = 0.0 kcal/mol, $\mu \left[C_{60}@ie\right]$ = 7.415 kcal/mol; $\mu \left[C_{60}@ee\right]$ = 13.492 kcal/mol). From a thermodynamic point of view, the complex $C_{60}@$*ii* was the most stable species, which was, moreover, the only complex expected at equilibrium. A question that is raised is whether complexation only takes place via the conformer *ii* or whether conversion to the *ii* conformer (to form $C_{X}@$*ii*) takes place after attachment of the fullerene. To investigate this point, we re-optimized the GFN2-xTB transition states for the interconversion between conformers of the buckycatcher with $C_{60}$ in between the pincer’s arms.

From a structural point of view, the effect of $C_{60}$ on the transition state is to further spread the pincer’s arms. In the transition state between *ii* and *ie*, $R_{P}$ increased by 1.05 Å, whereas for the other transition, the increase was 1.56 Å. The transitory arm was as flat as in the transition state without fullerene, and there was no bending whatsoever of the corannulene units in any structure.

When going from *ie* to *ii*, the fullerene stabilized the transition state by approximately 2 kcal/mol with respect to the free buckycatcher. $C_{60}$ thus favored the conversion of *ie* to *ii* by a factor of 20–30. Similar conclusions apply to the transition ee to ie, rendering these observations independent of the transition. Because of the significant stabilization that $C_{60}$ brings to *ii*, the interconversion of *ii* to *ie* was hindered by approximately 6 kcal/mol, which corresponded to a difference of four orders of magnitude in the relative kinetic constants (again, the same applied for the conversion of *ie* to *ee*). We conclude that the presence of the fullerene molecule in between the pincer’s arms favored the closing of the pincer’s arms to form structures with the arms pointing inwards.

The inclusion of toluene’s solvation effects barely had an impact on the relative energies, as it lowered all of the transition states by less than 0.2 kcal/mol.

### 2.5. Barriers for the Formation of Adducts with Fullerenes

To determine the barriers for forming adducts with fullerenes, we began with the optimization of the respective van der Waals adducts. Irrespective of the starting point, the system *ee* + $C_{60}$ always fell into the respective bound complex. We therefore expect that the formation of $C_{60}@$*ee* is a barrierless process. We note that this is easy to rationalize, strictly based on the structures of the species. For the other two conformers, we were able to isolate stable minima with the fullerene not encapsulated by the catcher. However, attempts to optimize transition states failed for both cases. Several approaches were tested, namely, the SADDLE algorithm of MOPAC [19] and xTB’s [20] reaction path finder [21]. The SADDLE calculations gave us geometries that were structurally and energetically very close to the bound complexes. The respective energy barriers would have been negative. Furthermore, tight optimizations of the transition state guesses always resulted in the bound complexes. Note that due to the reaction profile, an early transition state is to be expected, in accordance with the Hammond postulate. The late transition state predicted by MOPAC is thus not sensible.

To further investigate the issue, we decided to perform BOMD simulations (Figure 3) on the isolated catcher (baseline model, Appendix A) and on the van der Waals adduct of the catcher with $C_{60}$. In the beginning of the simulation, there was an increase in energy that was partly caused by (1) the energy averaging technique we employed to smear out the noise from the results and (2) also because the van der Waals adduct was not a point along the reaction pathway, though it was a minimum in the energy surface. As the simulation data showed, irrespective of the catcher conformer we considered, $C_{60}$ began by sliding sidewards from the respective minimum in energy. This promoted or at least gave the catcher enough time to further spread its arms so that the fullerene molecule could slide into its final position. A full cycle of opening and closing the catcher’s arms takes approximately 5–6 ps long, which is about the same time required for capturing the fullerene. From this, and because there is no bias (potential) used in the dynamical simulations, we expected the capturing of fullerenes to be a barrierless process.

During the dynamics of *ii* with $C_{60}$, we observed a change in conformation. This was because of the way the fullerene approached the buckycatcher. When sliding sidewards, the strong dispersion attraction between species makes the catcher open less from the side it is approached by $C_{60}$. Prior the formation of the complex, the catcher’s arm bends. A video based on these dynamical simulations is available in the Appendix A.

For comparison purposes, we simulated the complex $C_{60}@$*ee* for 350 ps, and no change in conformation took place (trajectory available in the Appendix A). Since the bending of the catcher’s arms in *ee* had a low barrier (the activation energy we calculated above was approximately 6 kcal/mol), we considered this as further support for the capturing of fullerenes to be a barrierless process.

### 2.6. The Energy Surface of Tetrachloroethane

A sketch of the potential energy surface for tetrachloroethane in its free form and within the arms of the buckycatcher is provided in Figure 4. Irrespective of its whereabouts, tetrachloroethane A (TCA) (the structure of conformers is given in Figure 4) was always the most stable conformer. The encapsulation by the buckycatcher led to a destabilization of tetrachloroethane B (TCB) relative to TCA; the effect was, however, minor at the GFN2-xTB level (at most 1 kcal/mol). The interconversion barrier between free-TCA and free-TCB was of 3–4 kcal/mol, depending on the direction of inversion. Curiously, the effects of the buckycatcher on this transition depended on the catcher’s conformation itself. Within *ee*, the transition state A2B was stabilized in gas, and the interconversion barrier was lowered due to the stabilization of the transition state. On the other hand, within *ie* or *ii*, the transition state was destabilized. This could be due to the flattening of the structure over the noncoplanar chlorine atoms.

The transition between the two forms of TCB was characterized by a barrier of approximately 6 kcal/mol. The larger barrier was justifiable by the coplanarity of all four chlorine atoms. Again, the presence of conformer *ee* had a stabilizing effect over this transition state, whereas *ii* and *ie* also made the interconversion barrier higher. The effects were, however, not as pronounced as for the transition A2B.

Because of the relative height of the interconversion barriers, the equilibrium between conformers was expected to be established rather quickly. The introduction of tetrachloroethane’s dielectric medium had a minor effect on the transition states of free tetrachloroethane. TCB was, however, stabilized. The barrier for forming TCB from TCA was lowered by 0.15 kcal/mol, though transforming TCB into TCA became slightly harder. On the other hand, the barrier between TCB was lowered by 0.19 kcal/mol. The solvation effects on the interconversion barriers inside the catcher were also dependent on the conformation of the buckycatcher. When tetrachloroethane was within *ee*’s arms, the barrier A2B increased by 2.053 kcal/mol, whereas the reverse of this transformation had a barrier larger by 2.937 kcal/mol. This was due to the stabilization of TCB@ee in comparison to free TCB, while the transition state was destabilized. Interconversion between conformers TCB was hindered by 3.768 kcal/mol, again for similar reasons. In the presence of *ii*, barriers were lowered by 2.765 (A2B), 1.406 (B2A) and 1.872 kcal/mol (B2B). Similar effects were observed for tetrachloroethane in *ie*, though the differences were milder. Thus, when in solution, the potential wells around each conformer of tetrachloroethane became more defined than in comparison to the gas phase. Though in gas, the presence of the corannulene pincer may lower the interconversion barriers between a few species, when in solution there is no meaningful acceleration of equilibration of species due to the presence of the buckycatcher.

We also investigated some of these barriers at the dispersion-corrected PM6 methods, and the general conclusions remained unchanged. We note, however, that the neglect of diatomic differential overlap (NDDO) methods penalize TCB over TCA and give this species a smaller contribution to the equilibrium.

### 2.7. Binding to Fullerenes in Tetrachloroethane

In this section, we study the binding to fullerenes in tetrachloroethane. We considered direct binding (D), where the solvent’s role is purely environmental (either with ALPB or the COSMO models) and a hybrid solvation model (E, for exchange binding). In the latter, one molecule of solvent was used in conjugation with implicit solvation. This is reasonable, because we found that the buckycatcher should form stable complexes with tetrachloroethane in solution [12]. The same would, however, not apply to toluene.

Much like the case of toluene, (implicit) solvation raised the Gibbs free energy for the formation of the adduct. PM6-D3H4X and PM6-D3H* were particularly adequate when mixed with ALPB, even if this model was parametrized for GFN2-xTB only: both the binding of $C_{60}$ and $C_{70}$ came close to their experimental values. The agreement between PM6-D3H* and the experimental data was not as good (as it was with PM6-D3H4X), which we attribute to the lack of repulsive dispersion in the geometry optimization. This goes together with the calculated energies at the PM6-D3H+ level. Our data show, furthermore, that using COSMO does not lead to such large solvation penalties for the binding process. This results in larger deviations with respect to the experimental data of at least 5 kcal/mol.

When moving to exchange-binding models, gas-phase enthalpies approached the experimental values, even though GFN2-xTB was still significantly off in comparison to the PM6-based methods. This was related to the over-binding effect between the pincer and fullerenes observed here for this model. Changes in entropy were also improved with respect to the direct-binding cases. The largest deviation in the experimental data was 4 cal/(K.mol), which, at room temperature, corresponded to a deviation of approximately 1 kcal/mol in Gibbs free energy.

When weighting enthalpy and entropy, we verified once more that PM6-based methods were closer to the experimental data than GFN2-xTB. This was exclusively due to the evaluation of enthalpies. Despite the missing bulk effect of the solvent, gas-phase Gibbs free energies were reasonably close to the experimental values. It is, therefore, tempting to consider that the main solvation effect for the binding of fullerenes in tetrachloroethane results from the exchange of the binding partner to the buckycatcher, with a small contribution from the dielectric constant of the medium.

Interestingly, PM6-D3H+ and PM6-D3H* yielded very similar results in the exchange reactions. This was because the three-body dispersion amounted to, at most, 0.5 kcal/mol. The smaller contribution of three-body terms in comparison to the direct-binding case shows that the hybrid model has the potential to reduce several errors stemming from the different models, particularly also higher-order dispersion. Because we analyzed gas-phase Gibbs free energies, we may assign the deviations, with respect to the experimental data, to the missing bulk effect of the solvent. Although not tested here, the proper inclusion of the Axilrod–Teller–Muto term in PM6-D3H* (i.e., with its effects also in gradients and geometry optimization) is expected to separate the results of these two models. Since this term adds repulsion between the atomic centers, we expect that energies, enthalpies and Gibbs energies increase, possibly leading to better agreement with the experimental data.

Introducing the correction from ALPB led to an increase in Gibbs free energies, typically retaining the accuracy of the gas-phase data. On the other hand, the COSMO model decreased Gibbs free energies, which negatively affected agreement with the experimental data.

Comparing these results with the ones in toluene (direct binding), we saw, overall, that semi-empirical methods with ALPB reproduced better the experimental data. Currently, we cannot truly point to whether the source of errors was the ALPB model itself or the inappropriateness of using benzene as a replacement for toluene. Note that in the present case, we also used the parametrization of a similar solvent as a replacement for inexistent parameters for tetrachloroethane. Nevertheless, even in the case of toluene, ALPB provided more reliable solvation effects than COSMO did, at least for the buckycatcher–fullerene system in these media.

## 3. Discussion

In this study, semi-empirical methods were used to investigate the binding of fullerenes to the buckycatcher. Particularly good agreement with experimental data was found on several occasions, which points to the suitability of these models to reproduce the chemistry of this system. We believe that this success, in comparison to higher-level DFT or ab initio methods, stemmed from a balanced accounting of dispersion. Note that unlike most density functionals, semi-empirical methods, particularly PM6, have no accounting whatsoever of dispersion. Therefore, D3 acts as a real and full correction for the lacking interactions. This goes hand in hand with the results of others using the Minnesota functionals [3,4]. Comparing PM6-D3H+ with PM6-D3H* stresses, furthermore, how the fine balancing of dispersion determines the success of the calculations. Though GFN2-xTB had dispersion terms in its Fockian, these were obtained by means of the D4 correction [22,23].

In general terms, we observed that PM6-D3H+ over-bound species, meaning that this method would significantly benefit from the proper introduction of three-body dispersion. Our estimates showed that Gibbs free energies in solution should then be of extremely high quality. PM6-D3H4X also showed quite good agreement with the reference data. Analysis of the weights given to the dispersion terms in PM6-D3H4X attests that this is no coincidence. The overall weight of two-body terms, $s_{6}$, took the value of 0.88. This is approximately 10% less than the $s_{6}$ in PM6-D3H+. The overall contribution of the estimated Axilrod–Teller–Muto was also approximately 10% of the two-body dispersion terms. This means that if repulsive three-body dispersion accounts to approximately 10% of the respective two-body attractive terms, then PM6-D3H4X and PM6-D3H* should be in good agreement. PM6-D3H4* would be in advantage when those fractions are unrealistic. On the other hand, our calculations revealed that GFN2-xTB also over binds the species. Further studies are, however, required to determine whether this is a weakness related to π-stacked systems.

Solvation also plays a very important role in the success of the simulations, and this study is no exception. Overall, COSMO seemed to be too weak to model the solvents herein considered. In some situations, ALPB overestimated the solvation effects, but this could have resulted from the misuse of parametrization. If we take, for instance, the PM6–D3H4X Gibbs free energies and apply the solvation penalty of ALPB for toluene, then the binding was weaker than the experimental one by approximately 2 kcal/mol. Nevertheless, the agreement obtained was better than most results reported in the literature. In the case of tetrachloroethane, the calculated Gibbs free energies in solution with ALPB matched better the data of Le et al. [15]. Since the binding energies at PM6–D3H4X agreed with the reference values of Sure and Grimme [7], the success of these calculations was also due to the solvation model and the rovibrational contributions to the thermodynamic functions. Though we did not, ourselves, calculate the COSMO-RS corrections, the use of these led to remarkable quality. The discussion by Sure and Grimme seemed, however, to indicate that this success might depend on the parametrization set utilized.

A hybrid solvation model was also considered, where the formation of an adduct with fullerene was transformed into an exchange reaction with the solvent. In this case, gas-phase thermodynamics approached the experimental ones which resulted in a better reflection of the behavior of the species in solution. Nevertheless, the final Gibbs free energies in solution were not of better quality than when using a direct-binding mode. We attempted to apply an identical (hybrid) model to account for the formation of the adducts in toluene. Although entropies improved significantly (−4.928 cal/(K.mol) for *ii* and −2.961 cal/(K.mol) for *ie*), the respective enthalpies worsened (ca. 14 kcal/mol), resulting in overall less qualitative Gibbs free energies. This is because the binding of toluene to the buckycatcher is not thermodynamically favorable [12], indicating that either there are other important effects not considered by the very simple model or the applicability is limited.

With regards to how the binding takes place, our simulations allowed us to assemble a quite vivid picture. Irrespective of compositions in solution and of conformer, the binding of $C_{60}$ to the buckycatcher seemed to be a barrierless process. Our dynamical simulations showed that the fullerene will tend to place itself next to the catcher and wait until the two corannulene units were far enough apart. As soon as that occurred, the fullerene jumped into its binding position. Because of the natural oscillatory movement exhibited by the buckycatcher, this was expected to be a rather quick process. Furthermore, due to the attractive π-stacking interactions, we expect that only one product was formed in this process, this being $C_{60}@$*ii*. This means that even if $C_{60}@$*ie* or $C_{60}@$*ee* are formed, the fullerene in between the corannulene units lowers the activation barriers in the direction of bending the corannulene rings inwards (towards the fullerene). These should, furthermore, be the rate-limiting steps in the binding, since barriers of approximately 6 kcal/mol must be crossed.

## 4. Materials and Methods

Calculations were performed using our newly developed C++ library, ULYSSES [24]. Geometry optimization was performed using the Broyden–Fletcher–Goldfarb–Shanno (BFGS) algorithm with the dogleg trust-region method in conjunction with convergence criteria of $10^{-8}$ $E_{h}$ for energies and $2.5\times 10^{-5}$ $E_{h}/a_{0}$ for gradients. The Hessian was approximated using the method of Lindh et al. [25]. Transition states were optimized using Baker’s RFO method [26] with more relaxed convergence criteria, namely, $7.5\times 10^{-4}$ $E_{h}/a_{0}$ for gradients. Numerical Hessians were used for geometry optimization of the latter, and the initial structures were obtained with the aid of Avogadro [27,28]. The final transition state structures were considered satisfactory when the gradient condition was satisfied and the only one imaginary frequency matched the case of free corannulene pincer. The optimized structures for the transition states were tested by adding/subtracting 15–20% of the Hessian’s eigenvector associated to the imaginary eigenvalue. We verified that further geometry optimization of these perturbed geometries would lead either to the reagent or to the products of the processes considered. Further details on how we obtained starting guesses for these structures are provided in the Appendix A.

The Hamiltonian of choice for optimizing geometries was always consistent with the method chosen for energy and Hessian evaluation. These were GFN2-xTB [14], PM6 [29], PM6-D3H4X [30,31,32] and PM6-D3H+ [33,34]. We wish to stress that in the D3H4X correction, there were hydrogen repulsion contributions, which were present in the simulations involving the corannulene pincer, meaning that D3H4X accounted for more than just attractive dispersion effects. For D3H+, there was only Grimme’s D3 correction, meaning that PM6-D3H+ was, in this case, equivalent to PM6-D3. Consequently, the two methods account differently for dispersion forces. When suitable, we corrected PM6-D3H+ with three-body dispersion. The Axilrod–Teller–Muto terms were, however, calculated using nonoptimal parameters. This means that we followed the previous approach of Grimme [30], and we also set the weights of the three-body terms to one ($s_{9}$ = 1). To distinguish from PM6-D3H+, we termed this method PM6-D3H*. We excluded PM7 [35] from the portfolio because its performance was lower for this type of system [36], at least when compared to PM6-D3H4. We stress that since PM7 already accounts for dispersion forces internally, extensions of this model with Grimme’s D3 or D4 corrections are not straightforward.

For the calculation of thermodynamic properties, we used the free-rotor/harmonic oscillator interpolation model of Grimme [5], which we extended for thermodynamic quantities other than entropies [24]. In our program, the interpolation frequency required by this model took the value of 75 cm^−1^. We also calculated properties using the traditional harmonic oscillator approximation as described in textbooks [37,38]. Because the harmonic oscillator fails for low-frequency internal modes—such as those characteristic of nonbonded aggregates—we only present data calculated with the FR/HO partition function. We comment, however, on the performance of the harmonic oscillator partition function when suitable.

Solvation Gibbs free energies at the GFN2-xTB level were estimated using the ALPB [39]. Note that this method was only parametrized for GFN2-xTB and, therefore, any solvation energy reported at the ALPB level was estimated based on the GFN2-xTB data (even if we mixed ALPB energies with other methods). Solvation effects were also calculated with the COSMO model [40] as available from MOPAC [19] in conjunction with PM6(-D3H4X). The solvent dielectric constant used in the COSMO/toluene calculations was 7.0, higher than the experimental dielectric constant of toluene. This is, however, in agreement with the parametrization of ALPB. The actual dielectric constant of toluene was also sporadically employed to verify that there was qualitatively no influence on the results. For tetrachloroethane, the solvent dielectric constant used was 8.42. Because ALPB was not parametrized for tetrachloroethane, the parametrization for dichloromethane was used instead. Solvation Gibbs free energies were always used with the respective data in the gas phase at 300 K. Equilibria in solution was discussed in conjunction with the FR/HO results. Simulations of the catcher or any other system in toluene used the parametrization for benzene, as we previously established that the parametrization for toluene might be at fault [12].

In some rare occasions, we were unable to obtain adduct geometries without imaginary vibrational frequencies, which might be extremely difficult to remove for very flexible systems. In such situations, we used the absolute value of the respective imaginary vibrational frequencies in the calculation of thermodynamic properties as suggested by Sure and Grimme [7]. In no case was a structure accepted with more than one imaginary frequency larger than 20i cm^−1^. Absolutely no vibrational frequency was disregarded in any part of this work.

The dynamics were run using the ULYSSES Born–Oppenheimer molecular dynamics (BOMD) module [41], which uses a semi-empirical quantum chemical method for energy and gradient (force) evaluation. Though we set the maximum simulation time to 100 ps, simulations were prematurely terminated as soon as the adducts were formed. The time step was set to 1 fs, and geometries for the construction of the trajectories were saved every 10 fs. Proton masses were rescaled to deuterium’s in order to allow for reasonably longer time steps. Integration of the equations of motion was conducted using the leapfrog algorithm [41]. Note that no sort of bias was used in the simulation.

For simplicity, conformational entropies in toluene were calculated considering that isomers of the buckycatcher are isoenergetic. This introduces deviations of, at most, 0.29 cal/(K.mol) in the final entropies (and were negligible for other thermodynamic properties). Due to the relative stability of the adducts with fullerenes and the low barrier of interconversion between these, we considered a single reaction product, $C_{X}@$*ii* (X = 60,70). This applied for the reactions in toluene and in tetrachloroethane.

Plots were created with Python’s matplotlib [42]. Figures of molecules were generated with UCSF Chimera, as developed by the Resource for Biocomputing, Visualization, and Informatics at the University of California, San Francisco, with support from NIH P41-GM103311 [43].

## 5. Conclusions

In this work, several semi-empirical methods were used to study the complexation of fullerenes to the buckycatcher in gas, toluene and tetrachloroethane. Solvation effects were introduced by means of COSMO, ALPB or a hybrid solvation model. Furthermore, we tested the effects of COSMO-RS based on results published in the literature. Overall, we observed that COSMO-RS may yield remarkable results. The ALPB offered a very good alternative, particularly when direct binding was studied (implicit solvation only). The inclusion of one explicit molecule of solvent showed the potential to bring gas-phase thermodynamics closer to the experimental values in solution. The data in solution were, however, not of improved quality.

Several methods and techniques implemented in our program ULYSSES were used to better understand the buckycatcher fullerene system. By means of BOMD simulations, we expect the formation of these complexes to take place in a barrierless process, determined solely by the rate of oscillations of the buckycatcher. Furthermore, we observed that irrespective of the conformer that binds to fullerene, the final product should always be the species $C_{60}@$*ii*. This is favored by thermodynamics and the simple fact that the fullerene molecule that sits between the two corannulene units favors the inward bending of the corannulene moieties to enhance π stacking.

## Figures and Tables

**Figure 1 molecules-27-03838-f001:**
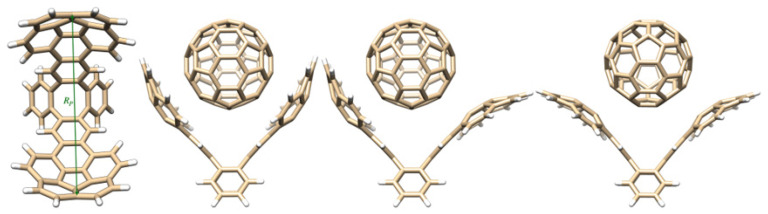
From left to right: definition of the $R_{P}$ parameter used to evaluate the width of opening for the buckycatcher and the structure of the buckycatcher complexes with $C_{60}$; complex formed with *ii*; adduct with *ie*; complex with *ee*.

**Figure 2 molecules-27-03838-f002:**
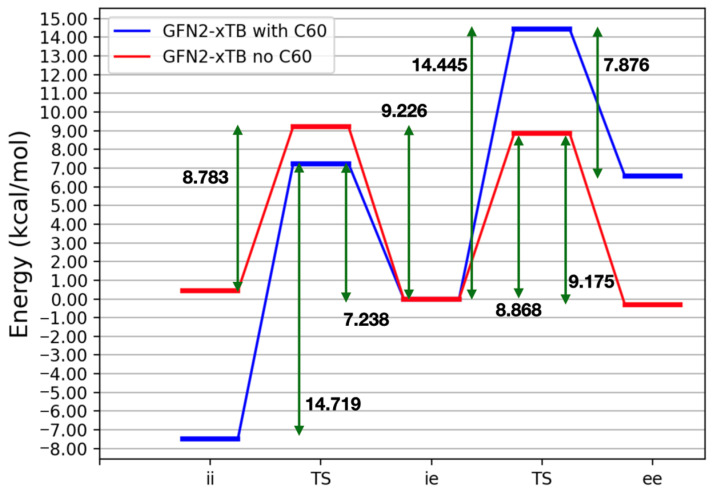
Energy (in kcal/mol) diagram for the interconversion of the corannulene pincer conformers with a molecule of $C_{60}$ within the pincer’s arms, according to GFN2-xTB. Only stationary points along the energy surface were calculated.

**Figure 3 molecules-27-03838-f003:**
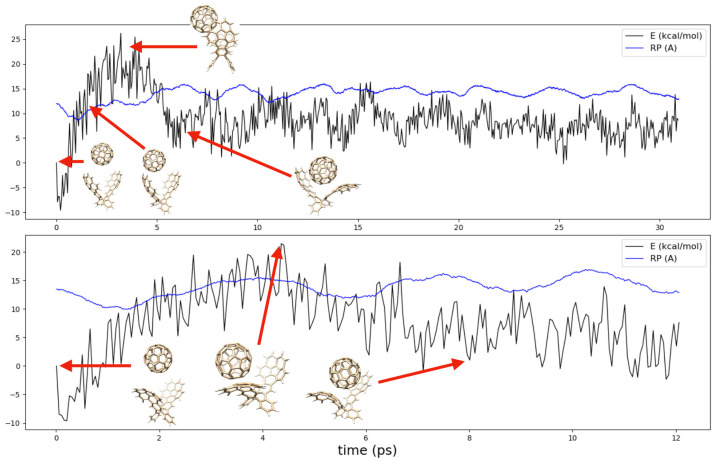
Dynamics of the catcher–$C_{60}$ van der Waals complexes.

**Figure 4 molecules-27-03838-f004:**
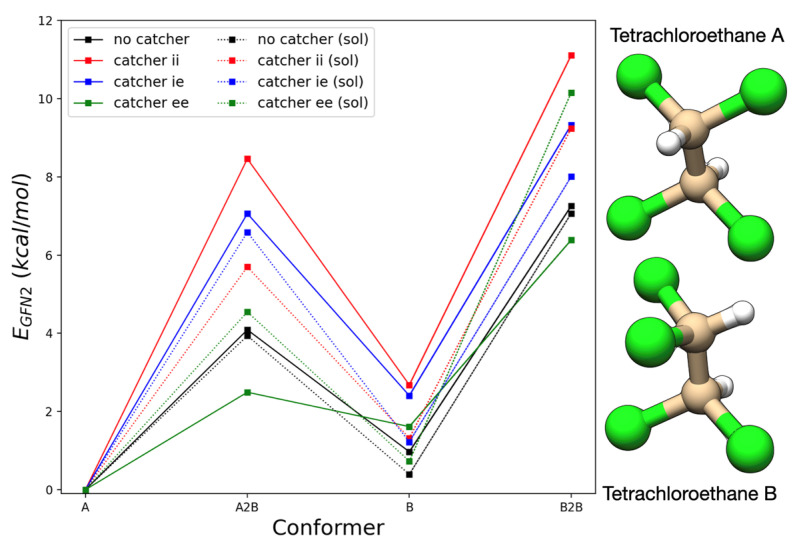
The energy surface of tetrachloroethane free and in between the arms of specific conformers of the buckycatcher according to GFN2-xTB. Relative energies are in kcal/mol.

**Table 1 molecules-27-03838-t001:** $R_{P}$ distances (in Å) or the several conformers of the corannulene pincer according to the four methods used in this study.

	PM6-D3H4X	PM6-D3H+	GFN2-xTB
*ii*-free	10.49	10.44	12.01
*ee*-free	12.29	12.56	13.46
*ie*-free	14.28	14.49	14.91
$C_{60}@$ *ii*	13.69	13.15	12.93
$C_{60}@$ *ee*	14.78	14.84	14.71
$C_{60}@$ *ie*	15.28	15.76	16.25

**Table 2 molecules-27-03838-t002:** Interaction energies for the buckycatcher fullerene complexes according to several semi-empirical methods. Interaction energies are in kcal/mol. Two sets of interaction energies are reported here for conformer *ii*: the main set corresponds to the method-specific interaction energies, based on the respective optimized structures; the second set corresponds to structures as reported in the S12L database [13] (in parenthesis).

	$C_{60}@ii$	$C_{60}@ie$	$C_{60}@ee$	$C_{70}@ii$	$C_{70}@ie$	$C_{70}@ee$
$\Delta E_{bind}^{PM6-D3H4X}$	28.062 (30.212)	22.189	16.115	30.397 (33.270)	24.249	17.808
$\Delta E_{bind}^{PM6-D3H+}$	31.156 (32.729)	24.501	17.638	33.456 (35.657)	26.530	19.409
$\Delta E_{bind}^{GFN2}$	39.623 (37.505)	31.392	25.130	41.240 (39.616)	33.703	26.369
Reference [7] ^1^	28.4	-	-	29.8	-	-

^1^ Permission to reuse data granted by ACS. Further reuse of this data must be directed to ACS. https://pubs.acs.org/doi/10.1021/acs.jctc.5b00296.

**Table 3 molecules-27-03838-t003:** Gibbs free energies (kcal/mol) for the formation of buckycatcher fullerene complexes according to several semi-empirical methods using several solvation models. Conformer-averaged data with conformational entropy were considered.

	$C_{60}@C_{60}H_{28}$	$C_{60}@C_{60}H_{28}$
$\Delta G_{PhMe}^{GFN2/ALPB}$	−12.867	−11.976
$\Delta G_{PhMe}^{PM6-D3H4X/ALPB}$	−1.888	−1.497
$\Delta G_{PhMe}^{PM6-D3H+/ALPB}$	−4.314	−3.801
$\Delta G_{PhMe}^{GFN2/COSMO}$	−21.398	−21.684
$\Delta G_{PhMe}^{PM6-D3H4X/COSMO}$	−10.419	−11.205
$\Delta G_{PhMe}^{PM6-D3H+/COSMO}$	−12.845	−13.509
$\Delta G_{PhMe}^{GFN2/COSMO-RS}$	−16.991	−16.940
$\Delta G_{PhMe}^{PM6-D3H4X/COSMO-RS}$	−6.012	−6.461
$\Delta G_{PhMe}^{PM6-D3H+/COSMO-RS}$	−8.438	−8.765
$\Delta G_{tol}^{exp}$ [15] ^1^	−4.77	−4.99

^1^ Permission to reuse data granted by ACS. Further reuse of this data must be directed to ACS. https://pubs.acs.org/doi/10.1021/jp5087152

**Table 4 molecules-27-03838-t004:** Thermodynamic data referring to the formation of the adducts between fullerenes and the corannulene pincer according to two different modes: direct binding (D) in which a molecule of the buckycatcher captures a fullerene; exchange binding where fullerenes replace caged solvent molecules. Enthalpies and Gibbs free energies are given in kcal/mol, whereas entropies are in units of cal/(K.mol). Experimental data are given for comparison purposes.

	$C_{60}\;D$	$C_{60}\;E$	$C_{70}\;D$	$C_{70}\;E$
$\Delta H_{gas}^{GFN2}$	−38.605	−20.353	−40.394	−22.167
$\Delta H_{gas}^{PM6-D3H4X}$	−27.670	−7.272	−29.571	−9.184
$\Delta H_{gas}^{PM6-D3H+}$	−30.527	−5.811	−28.583	−8.467
$\Delta H_{gas}^{PM6-D3H*}$	−26.950	−6.481	−28.583	−8.467
$\Delta H_{CH2Cl4}^{exp}$ [15] ^1^	−5.9		−5.3	
$\Delta S_{gas}^{GFN2}$	−50.796	−4.997	−54.594	−8.548
$\Delta S_{gas}^{PM6-D3H4X}$	−50.658	−2.273	−52.874	−4.485
$\Delta S_{gas}^{PM6-D3H+}$	−52.111	−1.570	−54.977	−4.433
$\Delta S_{gas}^{PM6-D3H*}$	−52.123	−1.990	−54.988	−4.854
$\Delta S_{CH2Cl4}^{exp}$ [15] ^1^	−6.0		−4.3	
$\Delta G_{gas}^{GFN2}$	−23.366	−18.854	−24.016	−19.603
$\Delta G_{gas}^{PM6-D3H4X}$	−12.473	−6.590	−13.708	−7.839
$\Delta G_{gas}^{PM6-D3H+}$	−14.894	−5.340	−16.009	−6.467
$\Delta G_{gas}^{PM6-D3H*}$	−11.314	−5.884	−12.086	−7.011
$\Delta G_{CH2Cl4}^{GFN2/ALPB}$	−15.064	−14.483	−14.067	−13.586
$\Delta G_{CH2Cl4}^{PM6-D3H4X/ALPB}$	−4.173	−3.014	−3.762	−2.616
$\Delta G_{CH2Cl4}^{PM6-D3H+/ALPB}$	−6.594	−1.715	−6.062	−1.195
$\Delta G_{CH2Cl4}^{PM6-D3H*/ALPB}$	−3.014	−2.048	−2.140	−1.528
$\Delta G_{CH2Cl4}^{GFN2/COSMO}$	−20.836	−20.367	−21.313	−20.945
$\Delta G_{CH2Cl4}^{PM6-D3H4X/COSMO}$	−9.968	−8.449	−11.031	−9.526
$\Delta G_{CH2Cl4}^{PM6-D3H+/COSMO}$	−12.390	−7.103	−13.332	−8.059
$\Delta G_{CH2Cl4}^{PM6-D3H*/COSMO}$	−8.805	−7.554	−9.406	−8.509
$\Delta G_{CH2Cl4}^{exp}$ [15] ^1^	−4.1		−4.0	

^1^ Permission to reuse data granted by ACS. Further reuse of this data must be directed to ACS. https://pubs.acs.org/doi/10.1021/jp5087152.

## Data Availability

All the data related to this manuscript (optimized geometries, frequencies, thermodynamic properties, input files, etc.) will be available in the GitLab repository (https://gitlab.com/siriius/buckycatcherrevisited.git) upon publication of this manuscript.

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
