# Peer review of "How to Catch the Ball: Fullerene Binding to the Corannulene Pincer"

_molecules, 2022, doi:10.3390/molecules27123838_

Round 1
Reviewer 1 Report
Dear Authors,
I had the pleasure of reading your manuscript and I find it highly suitable for this journal. Your work is of interest for at least two reasons. Firstly, you have developed a new computational tool (ULYSSES), and second, you have applied it to study interesting and important phenomena.
From the fundamental and technical aspects of your manuscript, there is nothing serious that would require modifications of your manuscript. Therefore, I am recommending it to be accepted for publication in this journal.
All the best
Author Response
Dear Reviewer,
thank you so much for your kind and encouraging comments.
Best wishes,
Filipe Menezes
Reviewer 2 Report
The submitted manuscript is very interesting and of high quality, in terms of both scientific soundness and importance of the results. I have only some minor questions and suggestions.
Line 457-458, in the future studies I would recommend using the LBFGS algorithm as it is generally recommended, especially for large systems. It implements a universal sparse preconditioner that accelerates geometry optimization.
Lines 468-469, why the Authors have not considered PM7? Of course with suitable dispersion corrections.
I understand that the modelled systems are large which fully justifies the use of semi-empirical methods, especially for the MD part. However, I am curious whether the Author did not consider using the DFT, at least for single point calculations? I have a very good experience with MBD correction.
In the introduction some attention should be paid to describe the ii and ee conformations and their energy barrier. Is it possible, at normal T, to observe such transitions (ii=ee)? Was this phenomenon studied experimentally or computationally?
Figure 2. How exactly have the Authors obtained the transition states? And have you performed any TS confirmation?
Author Response
Please see the attached file.
Thanks in advance
